# Continuous Wear of Night and Day Orthosis Is a Key Factor for Improvement of Fixed Equinus Deformity after the Transverse Vulpius Procedure

**DOI:** 10.3390/children9020209

**Published:** 2022-02-06

**Authors:** Mathis Wegner, Katharina I. Koyro, Louisa Kosegarten, Anna Kathrin Hell, Heiko M. Lorenz, Volker Diedrichs, Sebastian Lippross

**Affiliations:** 1Department of Trauma and Orthopaedic Surgery, University Medical Center of Schleswig-Holstein, Campus Kiel, 24105 Kiel, Germany; louisa-kosegarten@web.de (L.K.); sebastian.lippross@uksh.de (S.L.); 2Department of Plastic, Aesthetic, Hand and Reconstructive Surgery, Burn Centre, Medical School Hannover, 30625 Hannover, Germany; koyro.katharina@mh-hannover.de; 3Department of Orthopaedics, Trauma and Plastic Surgery, Paediatric Orthopaedics, University Medicine Goettingen, 37099 Göttingen, Germany; anna.hell@med.uni-goettingen.de (A.K.H.); heiko.lorenz@kwm-klinikum.de (H.M.L.); 4Lubinus Clinicum Kiel, 24106 Kiel, Germany; kinderorthopaedie-diedrichs@lubinus-stiftung.de

**Keywords:** equinus deformity, gastrocnemius release, gastrocnemius–soleus aponeurotic recession

## Abstract

*Background:* Equinus foot deformity is secondary to either spasticity or contracture of the gastrocnemius–soleus complex. The plantar flexion is basically treated conservatively; several different surgical methods have been discussed. This paper focuses on the improvement of passive ankle dorsiflexion after a transverse Vulpius procedure in equinus foot deformity. Additionally, the influence of consequent postoperative wear of orthosis on the improvement of ankle range of motion was investigated. *Methods:* In total, 41 patients with neuromuscular impairment and 59 equinus feet deformities were surgically treated by using a transverse Vulpius procedure. A total of 19 female patients and 22 male patients with a mean age at surgery of 10.18 years (2 to 31) were included. Mean follow-up took place 12.26 ± 7.95 months after surgery. Passive ankle dorsiflexion was measured and subjective patients’ satisfaction was assessed. *Results:* Range of motion, measured as the maximum of passive ankle joint dorsiflexion, improved significantly from −8° ± 5.9° to 11.1° ± 6.7° directly after surgery to 16.2° ± 10.7° at follow-up. The improvement of passive ankle dorsiflexion was significantly associated with the continuous wearing of night and day orthosis (*p* = 0.0045). Patient subjective satisfaction was very high. *Conclusion:* A transverse Vulpius procedure for aponeurotic gastrocnemius and soleus muscle lengthening of equinus foot deformity resulted in a significant improvement of passive ankle dorsiflexion. Positive surgical results correlated to a continuous use of orthotic devices.

## 1. Introduction

Equinus foot deformity is secondary to either spasticity or contracture of the gastrocnemius–soleus complex (GSR) [1]. This deformity causes the ankle to plantar flex, meaning that children walk on either their forefoot or just their toes [2]. As a symptom of several neurological disorders, it is one of the most common and disabling pathologies in children with spastic hemiplegic and diplegic cerebral palsy [3]. Even though an ankle dorsiflexion greater than 10° in the extended knee position is necessary for a physiological gait pattern [4,5], ankle range of motion represents a key kinematic factor for an efficient gait in patients with equinus foot deformity [6]. The interplay of spasticity, joint contractures and tone results in significant gait challenges, such as increased forefoot load, decrease in stance stability, disturbed swing phase and osseus deformities [2,6]. Especially in childhood, dynamic equinus foot deformity is common. If not targeted, fixed equinus and secondary osseus deformities may develop, altering individual gait pattern on different joint levels [7,8,9].

Treatment of dynamic equinus foot deformity is basically conservative. Improvements are achieved through repetitive stretching, short-term casting, ankle–foot orthoses or night splints, physical therapy and botulinum toxin injection [10]. After unsuccessful conservative treatment or progress deformity, surgical release may be discussed [7,11,12].

Several surgical methods for treating equinus foot deformity have been proposed in the literature [11,13,14,15,16,17]. The transverse gastrocsoleus Vulpius recession (GSR) describes an intramuscular aponeurotic lengthening of the triceps surae muscle in the deep interval between the gastrocnemius muscle and the soleus muscle [18]. The goal of this procedure is to increase ankle dorsiflexion and movement patterns when ankle movement is restricted by a contracted gastrocnemius muscle, contracted soleus muscle [1]. The gastrocnemius–soleus complex (GSC) is a group of three muscles that originate from the distal femur and the proximal tibia and inserts at the dorsal calcaneus, thereby spanning two joints. Described by Vulpius and Stoffel 1924 [19], the muscle–tendon unit of the calf is anatomically divided into three zones: Zone 1, extending from the origin of the muscle to the distal fibers of the medial belly; Zone 2, extending from the distal extent of the medial gastrocnemius muscle belly to the most distal fibers of the soleus; and Zone 3, the Achilles tendon (Figure 1) [12]. The lengthening of the GSC can either be complete, addressing the soleus and gastrocnemius, or partial, only involving the superficially lying gastrocnemius. The Silverskjiöld test helps to determine whether the shortening of the soleus is causative for reduced dorsiflexion and to identify the appropriate treatment method [20].

In the described patient population, the aponeurotic lengthening of the gastrocnemius and the soleus muscle was performed as a transverse Vulpius GSR [18]. Using GSR, satisfactory mid- and long-term results concerning ankle mobility and physiological gait pattern have been reported [11,14,18,21,22].

This paper focuses on the improvement in passive ankle dorsiflexion after a transverse Vulpius procedure in equinus foot deformity, as well as other postoperative factors influencing the improvement of ankle range of motion after aponeurotic lengthening of the gastrocnemius and the soleus muscle.

## 2. Materials and Methods

In a retrospective single-center evaluation study, 41 patients (22 male, 19 female) with surgically treated equinus foot deformity were identified and met the inclusion criteria. Patients undergoing unilateral or bilateral transverse Vulpius GSR for equinus gait with different underlying pathologies between 2017 and 2019 at the Medical Center Schleswig Holstein, Campus Kiel, were included. Patient demographics are described in Table 1. Exclusion criteria were a previous selective dorsal rhizotomy, the use of an intrathecal Baclofen pump and botulinum toxin injections within the last six months.

The study complied with the ethical guidelines for human studies, approval was given by the local ethics committee of the medical faculty of the University of Kiel on 30 January 2019 (No. D410/19) and informed consent was obtained from all patients and/or their legal caregivers.

In 41 patients, 59 equinus feet were surgically treated performing a transverse Vulpius gastrocsoleus procedure [18]. The gastrocnemius–soleus intramuscular aponeurotic recession (GSR) was performed through a longitudinal incision at the distal one-third of the soleus muscle belly in the interval between gastrocnemius and soleus muscle [18,19]. Sparing the greater saphenous vein and the saphenous nerve, the conjoined aponeurosis was exposed. The entire aponeurosis of the triceps surae, the intermuscular septum and the plantaris muscle tendon were incised transversely from medial to lateral under direct vision (Figure 1). Placing the knee in an extended position, a dorsiflexion stretch was applied, aiming for 10° dorsiflexion. After adequate dorsiflexion was noted and primary wound closure was performed, a below-the-knee cast in a plantigrade position was applied to secure dorsiflexion.

A knee immobilization orthosis was applied to secure the extension during the night. Postoperative care allowed immediate mobilization and weight-bearing to be tolerated to prevent muscle atrophy. After three to four days, the cast was replaced by an ankle-immobilizing orthosis (Figure 2) for the daytime and nighttime. The orthoses were customized individually by a trained orthopedic technician in the provision of pediatric aids. The orthosis was adapted to the range of motion of the ankle and, if necessary, to the individual remaining deformity. Regular examinations of the fit were necessary to prevent pressure marks. The orthoses were designed to reduce plantar flexion at the ankle during walking using a quenching mechanism with return spring. Adjustment of the return spring mechanism was possible. To prevent shortening of the gastrosoleus complex, an additional knee-immobilizing splint was prescribed for nighttime wear. All patients were encouraged to wear both orthoses seven days/nights a week for 12 months.

The range of motion of the ankle joint was documented prior to surgery, in the operation room, after surgical intervention and during follow-up. The exams were performed by the same two investigators. The ankle range of motion was measured using a standard orthopedic goniometer. A visual assessment of the gait pattern was conducted. A questionnaire focusing on subjective satisfaction and the current treatment regime was filled out at follow-up. Patient charts were reviewed for complication data.

Statistical analysis was performed using the unpaired two-way Students’ *t*-test or one-way analysis of variance (one-way ANOVA). Values given in the text are mean +/− standard deviation. Statistical significance was defined with levels as *p* < 0.05 (*), *p* < 0.01 (**) and *p* < 0.001 (***).

## 3. Results

In 41 patients with neuromuscular impairment, 59 equinus feet deformities were surgically treated by using the transverse Vulpius procedure. A total of 19 female patients and 22 male patients were included in our study. The mean age at surgery was 10.18 years (2 to 31). The mean follow-up took place 12.26 ± 7.95 months after surgery. All patients received the same surgical and post-surgical treatment.

Range of motion, measured as the maximum of passive ankle joint dorsiflexion, improved significantly directly after surgery and at follow-up (Figure 3).

Preoperative ankle dorsiflexion was -8° ± 5.9° and increased significantly to 11.1° ± 6.7° immediately after surgery. At follow-up (12.26 ± 7.95 months after surgery), an additional improvement to 16.2° ± 10.7° could be noted. (** *p* < 0.007; *** *p* < 0.0001, **** *p* < 0.0001). Data were analyzed by one-way analysis of variance and unpaired Students’ *t*-test.

Patients’ subjective satisfaction concerning strength in the lower limbs, range of motion and gait pattern showed a subjective improvement. Subjective gait pattern improvement was noted in 57 feet (40 patients). One ambulatory child was not satisfied with the surgical result (Table 2).

A subjective assessment of gait revealed a physiological gait pattern with heel-to-toe movement in *n* = 44 (74.6%) feet at follow-up. Pathological gait pattern persisted in *n* = 10 (16.9%) feet. A pathological gait pattern without heel contact was observed in *n* = 5 feet (8.5%). Gait analysis was omitted in *n* = 5 (3 patients) because the patients were unable to walk. In these five feet, no recurrent equinus foot deformities existed. Overall, there was only one recurrence, which was due to dorsiflexion of less than 0°.

To analyze the dependent factors on ankle dorsiflexion after performing an aponeurotic gastrocnemius and soleus recession, patients were divided into subgroups concerning their post-surgical use of orthosis. In 45.7% of surgically treated feet (*n* = 27 feet; *n* = 19 patients), orthoses were regularly used day and night, while in 35.6% of surgically treated feet (*n* = 21; *n* = 14 patients), feet orthoses were used either day or night. A total of 18.6% (*n* = 11; *n* = 8 patients) reported sporadic, irregular use. At follow-up, the first group presented with a dorsiflexion of 20° ± 10.9°, the second with 15.1° ± 9.6°, and the third with 8.9°± 8.2°. There was a significant difference between groups 1 and 3 (Figure 4).

At follow-up (12.26 ± 7.95 months after surgery), there was an average dorsiflexion of 20° ± 10.7° in children (*n* = 27 feet, *n* = 19 patients) who wore orthoses day and night 15.1° ± 9.6° if orthoses were applied either day or night (*n* = 21 feet; *n* = 14 patients) and 8.9° ± 8.2° (*n* = 11 feet, *n* = 8 patients) if orthoses were worn irregularly. There was a significant difference between groups 1 and 3 (***p* < 0.05). Data were analyzed by unpaired students’ two-way *t*-test.

The administration and frequency of physiotherapy was analyzed as another confounding factor on ankle dorsiflexion. Out of the 59 operated feet, *n* = 11 feet (18.6%) did not receive physiotherapy after surgery, *n* = 41 (69.5%) received physiotherapy once or twice a week, and *n* = 7 (11.9%) two times or more. At follow-up, the average values of dorsiflexion did not show significant differences (16.6° vs. 14.3° vs. 15.9° vs. 21°) (Figure 5).

At follow-up (12.26 ± 7.95 months), there was no significant difference in ankle dorsiflexion (16.6° vs. 14.3° vs. 15.9° vs. 21°) in feet that received no physiotherapy at all, once, twice or at least three times a week (n.s. = not significant). Data were analyzed by one-way analysis of variance and unpaired Students’ *t*-test.

During surgery, no complications occurred. Complications related to surgery were reported in 25 feet. In *n* = 8 feet, mild postoperative pain was reported. Mild postoperative paraesthesia was reported in *n* = 7 feet. *n* = 56 feet were included. Due to young age, *n* = 3 feet could not participate.

## 4. Discussion

For the surgical treatment of equinus foot deformity in children and adults, several surgical techniques have been described, mainly focusing on restoring length through surgery of the Achilles tendon, the gastrocnemius and/or soleus muscle [11,13,14,15,16,17]. There are varying opinions with respect to the appropriate surgical treatment method [11,14,23].

The aim of every calf-lengthening procedure is to achieve an appropriate length and strength of the GSC in patients, but to prevent over- or undercorrection resulting in gait impairment or revision surgery. Another aim is to maintain achieved correction after surgery. With ankle range of motion representing a key kinematic factor for an efficient gait in patients with equinus foot deformity [6], our results concerning post-surgical ankle dorsiflexion are excellent. Dorsiflexion improved initially from −8° ± 5.9° over 11.1° ± 6.7° directly postoperative to 16.2° ± 10.7° at follow-up. These data are in line with the effect that other studies describe using the same GSR procedure [18]. Several authors using different surgical procedures for equinus foot deformity reported satisfactory short-, mid- and long-term results as well [7,11,14].

Our demographics show a very heterogenous study cohort, containing patients suffering from infantile cerebral palsy, idiopathic equinus foot deformity, clubfoot deformity and cavus–equinus foot deformity, indicating that the transvers Vulpius GSR represents an excellent surgical procedure in different underlying pathologies. Compared to other studies treating equinus foot deformities, patient cohorts only include children with spastic diplegic cerebral palsy [11,14,18].

A calcaneus is hyperdorsiflexion of the ankle and is seen particularly as a result of overcorrection in surgery, causing a crouch gait and eventually loss of walking ability [2,11,14,23]. The chance of calcaneus occurring after isolated lengthening of the calf muscles in children with hemiplegic and diplegic equinus varies depending on different factors [2]. In our patients, 17% had a post-surgical impairment of gait, even though subjective post-surgical patient satisfaction in our cohort was very high. In a study including 134 children and 195 procedures, Borton et al. (2001) reported an incidence rate of calcaneus of 36%. Dreher et al. (2012) found that the chance of overcorrection and calcaneal impairment of gait was substantially lower after GSR (30%) than after a level 3 Achilles tendon lengthening (35%) according to Baumann and Koch [17]. In their study, overcorrection was observed in 9% of patients at short-term follow-up and 10% had calcaneal gait impairment at final follow-up. In a study of patients with diplegic cerebral palsy who were treated with GSR described by Baumann and Koch, Saraph et al. (2000) reported that no patients had overcorrection, which is in line with our results even though we used a different surgical approach.

Despite favorable results after surgery, only a few studies focus on post-surgical treatment and its influence on range of motion and recurrency of equinus [24]. Our data showed a positive correlation of ankle dorsiflexion and orthotic treatment day and night as opposed to irregular use of orthoses. By emphasizing orthotic treatment day and night during follow-up, ankle dorsiflexion significantly improved from 11.1° ± 6.7° directly post-surgery to 16.2° ± 10.7° at follow-up, with one patient having recurrent equinus. Orthotic treatment, as described by other authors, was inconsistent regarding the time and extent of post-surgical orthotic treatment [6,7,11,14,18]. Ranging from 0% to 48% [11], the prevalence of recurrent equinus has been attributed to different surgical approaches [22], age at the time of surgery, gender and skeletal maturity [6,24]. Our data identify the extent of post-surgical orthotic treatment as a possible confounder for recurrent equinus.

Physiotherapy itself may be an effective treatment for equinus foot deformity. In this study, all patients received physiotherapy as part of a conservative treatment regime prior to surgery, but only 83% continued after surgery. However, there was no correlation between physiotherapy and range of motion in this investigation. In the literature, this aspect is rarely mentioned and not investigated as a confounding variable in the described studies [7,11,14,18].

Despite these interesting results, there are several limitations to this study. A standardized 3D gait analysis, as performed in other studies [11,18], would have been favorable for the gait pattern assessment. Video and clinical evaluations, as performed in this investigation, are clearly inferior to standardized 3D evaluations. A full gait analysis would also provide information about the condition of the knee and hip. Additionally, a standardized questionnaire prior to surgery and at follow-up would have led to a more conclusive comparison. Even though post-surgical treatment regimens were thoroughly investigated, it cannot be determined whether patients and parents were completely honest about the use of orthotic devices. To investigate the influence of physiotherapy on equinus feet, a standardized protocol and standardized extent of physiotherapeutic treatment is needed. Our physiotherapeutic protocol aimed to restore the function lost through surgery directly after surgery. Later, the protocol aimed to conserve the gain in range of motion, balance and function. However, a detailed assessment of the compliance, exact physiotherapeutic treatments and the influence of these treatments on dorsiflexion were missing. Finally, a longer follow-up duration would have been favorable.

## 5. Conclusions

In conclusion, a transverse Vulpius procedure for aponeurotic gastrocnemius and soleus muscle lengthening of equinus foot deformity in children resulted in significant improvement of ankle dorsiflexion. Positive surgical results correlated with the continuous use of orthotic devices.

## Figures and Tables

**Figure 1 children-09-00209-f001:**
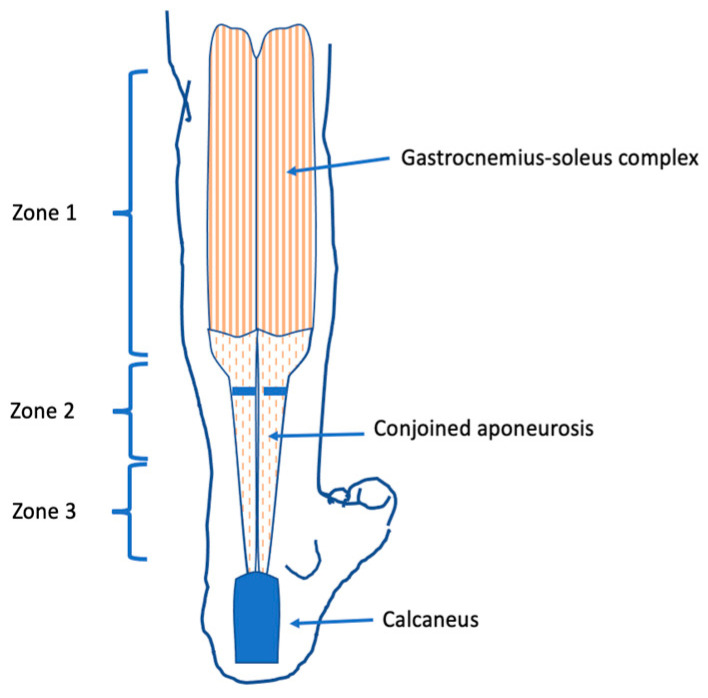
Line drawing transversely in Zone 2, showing the location of the transverse Vulpius gatrocsoleus recession.

**Figure 2 children-09-00209-f002:**
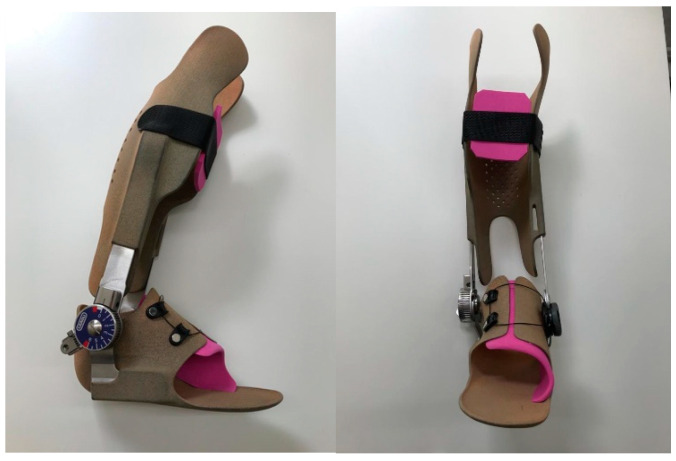
Images of an ankle immobilizing orthosis.

**Figure 3 children-09-00209-f003:**
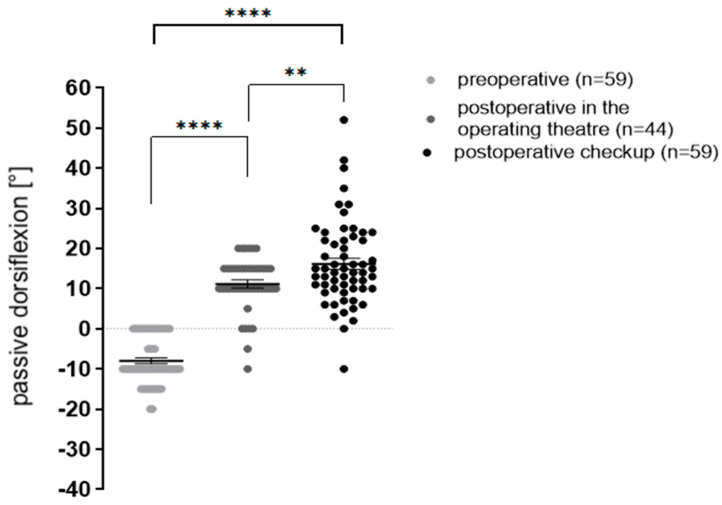
Dorsiflexion before, directly after surgery and at follow-up (** *p* < 0.007; **** *p* < 0.0001).

**Figure 4 children-09-00209-f004:**
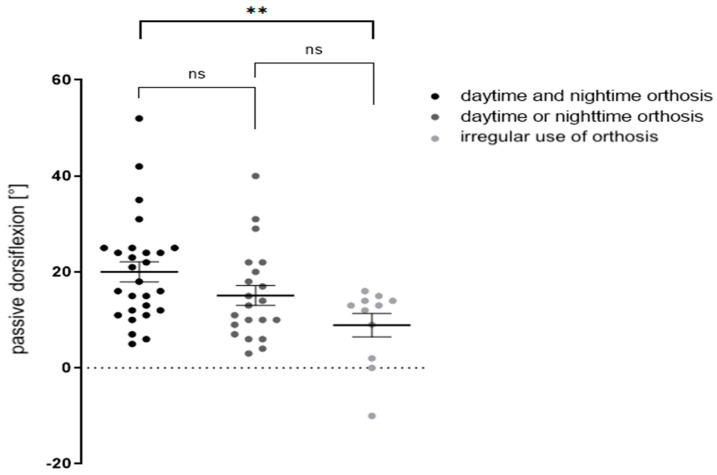
Impact of regular use of orthotic devices on passive ankle dorsiflexion at follow-up (** *p* < 0.05).

**Figure 5 children-09-00209-f005:**
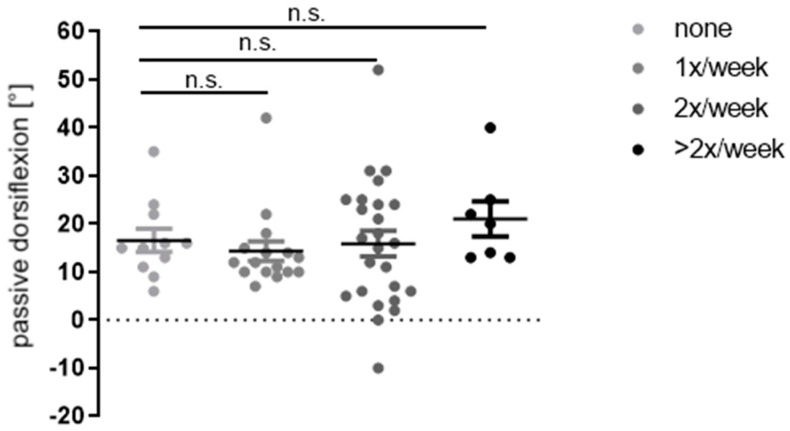
Influence of administration and frequency of physiotherapy on postoperative passive dorsiflexion.

**Table 1 children-09-00209-t001:** Patient demographics.

Variables	Value Patients (Feet)
Patients Surgically treated feet	41 59
Age at surgery	11.18 (5.99) median 10.5
Primary diagnosis Infantil cerebral palsy Idiopathic Clubfoot Cavus-equinus Other	36 feet 13 feet 5 feet 3 feet 2 feet
Ambulatory status Walking Wheelchair with ability to stand	39 (54) 2 (4)
Prior treatment to surgery Physiotherapy Intermittent casting Ankle–foot orthosis Night-splinting	30 (47) 39 (56) 36 (51) 36 (51)

**Table 2 children-09-00209-t002:** Subjective satisfaction at follow-up.

	Extremely Satisfied [%]	Satisfied [%]	Neutral [%]	Dissatisfied [%]
Strength *n* = 59	91.5	6.8	1.7	0
Range of motion *n* = 59	84.7	11.9	3.4	0
Gait pattern *n* = 54	68.5	22.2	7.4	1.9

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
