# Peer review of "Continuous Wear of Night and Day Orthosis Is a Key Factor for Improvement of Fixed Equinus Deformity after the Transverse Vulpius Procedure"

_children, 2022, doi:10.3390/children9020209_

Round 1

Reviewer 1 Report

Major comments

This study used the modified Vulpius GSR for equinus foot deformity and a measurement of passive ankle dorsiflexion after wearing the orthosis day and night was performed. The results showed a great improvement for the patients. Here are some comments:

  1. This study may disclose a figure about the modified Vulpius GSR in the description of “ Materials and Methods”.
  2. An ankle immobilizing orthosis was applied to the post-operation care. Why did you use this kind of orthosis? The features of the orthosis which is helpful for the care should be discussed.
  3. This study should compare the prior researches by that: a. the difference of (modified) GSR without wearing orthoses, and b. the difference of (modified) GSR using other post-operative care. The comparison had better reveal in a table.
  4. Line 192: “represents an excellent surgical procedure in different pathologies” If so, the different pathologies should be mentioned in “ Materials and Methods”, and the quantity of different patients may be requested for the significant statistical analysis.

Minor comments

Abstract

Line 29: This study worked well. It is just suggested that “Excellent” surgical results should be described objectively and humbly.

Conclusions

Line 240: the same comment with Line 29.

Reviewer 2 Report

The authors present an investigation into the passive ankle dorsiflexion achieved post surgically with the modified transverse Vulpius procedure for patients with equinus deformity of the ankle as well as postsurgical patient satisfaction.  Surgery resulted in improvement from -8° ± 5.9° 24 to 11.1° ± 6.7° and at follow up was found to be 16.2° ± 10.7°.  All patients were prescribed a stretching orthosis postsurgically, some also received physical therapy.

Current treatment methods of toe-walking/ankle equinus include surgery, repetitive stretching, short-term casting, ankle-foot orthoses or night splints, physical therapy and botulinum toxin injection (surgery only if conservative treatment fails).

The postoperative results of this study show that daytime and nighttime use of a stretching orthosis is more effective than part-time or inconsistent wear of the orthosis.  No statistical difference was found between groups who did/did not receive physiotherapy or among those within the physiotherapy group who received treatment at the rate of 1xWeek, 2xWeek, or >2xWeek. Only 1 patient was unhappy about the surgical outcome; the vast majority showed postsurgical positive outcomes in gait pattern improvement.

____

Detailed Manuscript Notes: 

In lines 59-63, I do not see a reason to describe the different zones mentioned by Vulpius and Stoffel, unless this is intended to be referenced later in the manuscript.

In the introduction, please describe the difference between the modified transverse Vulpius procedure and the transverse Vulpius procedure.

In the methods section, please explain how the ankle range of motion was measured.

Table 1 – define “ICP” within table or within table footnote.

The sentence in lines 119-121 are understandable, but awkwardly worded.  Please consider rewording this sentence.  In particular, the last part of the sentence is a bit confusing “…there was an additional improvement from initially -8° ± 5.9°, over 11.1° ± 120 6.7° to 16.2° ± 10.7° of ankle joint dorsiflexion.”  Since this information is repeated in the next paragraph, this entire line could be eliminated.

Figure 2 – why are some data points shown as lines and others as dots (when looking horizontally)?

Table 2 – for Range of Motion, the %’s add up to 101.7%.  Please double check numbers here.

Line 138-139: I think a word is missing here.  Did you intend to write “Overall, there was only one  recurrence which was due to dorsiflexion of less than 0°.”

Lines 142-144: This reports on 98.2% of patients with 58 feet on 40 patients.  Was someone not included in the follow up?  Please double-check reporting.  If anyone was excluded, explain why the exclusion.

Figure 3 – I don’t believe “unregularly” is a word, may be replaced with “irregular”

Line 151: change “15,1” to “15.1”

Line 152: change “8,9” to “8.9”

Lines 150-155: reports on 59 feet in 40 patients.  Please double-check numbers.

Line 160: please correct “16,6° vs.14,3° vs.15,9°” to “16.6° vs. 14.3° vs.15.9°”

In limitations section, please put more detail on limitations of physical therapy assessment.  For example, home stretching is typically prescribed with physical therapy but I do not see any assessments as to the compliance of home-stretching between sessions.

Is there existing literature to compare your post-operative and post-bracing results to non-braced results of the same modified transverse Vulpius procedure?  If so, it would be interesting to see the comparison since all patients in this cohort were instructed to wear the orthosis (although there were compliant and non-compliant patients).

____

Overall, this is a nicely written study with only minor content edits needed to improve readability and clarity.  Some reported statistics do need to be rechecked as noted above; recommend rechecking all numbers prior to publication.
